# Debiased Bayesian inference for average treatment effects

**Kolyan Ray**
Department of Mathematics
King's College London
kolyan.ray@kcl.ac.uk

**Botond Szabó**
Mathematical Institute
Leiden University
b.t.szabo@math.leidenuniv.nl

## Abstract

Bayesian approaches have become increasingly popular in causal inference problems due to their conceptual simplicity, excellent performance and in-built uncertainty quantification ('posterior credible sets'). We investigate Bayesian inference for average treatment effects from observational data, which is a challenging problem due to the missing counterfactuals and selection bias. Working in the standard potential outcomes framework, we propose a data-driven modification to an arbitrary (nonparametric) prior based on the propensity score that corrects for the first-order posterior bias, thereby improving performance. We illustrate our method for Gaussian process (GP) priors using (semi-)synthetic data. Our experiments demonstrate significant improvement in both estimation accuracy and uncertainty quantification compared to the unmodified GP, rendering our approach highly competitive with the state-of-the-art.

## 1 Introduction

Inferring the causal effect of a treatment or condition is an important problem in many applications, such as healthcare [11, 17, 38], education [20], economics [16], marketing [6] and survey sampling [13] amongst others. While carefully designed experiments are the gold standard for measuring causal effects, these are often impractical due to ethical, financial or time-constraints. For example, when evaluating the effectiveness of a new medicine it may not be ethically feasible to randomly assign a patient to a particular treatment irrespective of their particular circumstances. An alternative is to use *observational data* which, while typically much easier to obtain, requires careful analysis.

A common framework for causal inference is the *potential outcomes* setup [19], where every individual possesses two 'potential outcomes' corresponding to the individual's outcomes with and without treatment. For every subject in the observation cohort we thus observe only one of these two outcomes and not the 'missing' *counterfactual* outcome, without which we cannot observe the true treatment effect. This problem differs from standard supervised learning in that we must thus account for the missing counterfactuals, which is the well-known missing data problem in causal inference.

A further complication is that in practice, particularly in observational studies, individuals are often assigned treatments in a biased manner [36] so that a simple comparison of the two groups may be misleading. A common way to deal with selection bias is to measure features, called *confounders*, that are believed to influence both the treatment assignment and outcomes. The discrepancy in feature distributions for the treated and control subject groups can be expressed via the *propensity score*, which is then used to apply a correction to the estimate. Under the assumption of *unconfoundedness*, namely that the treatment assignment and outcome are conditionally independent given the features, one can then identify the causal effect. Widely used methods include propensity score matching [32, 34, 36] and double robust methods [5, 31, 33].

In recent years, Bayesian methods have become increasingly popular for causal inference due to their excellent performance, for example Gaussian processes [1–4, 11, 23, 38] and BART [13, 15, 17, 18, 20, 35] amongst other priors [6]. Apart from excellent estimation precision, advantages of the Bayesian approach are its conceptual simplicity, ability to incorporate prior knowledge and access to uncertainty quantification via posterior credible sets.

In this work we are interested in Bayesian inference for the *(population) average treatment effect* (ATE) of a causal intervention, which is relevant when policy makers are interested in evaluating whether to apply a single intervention to the entire population. This may be the case when one no longer observes feature measurements of new individuals outside the dataset. This problem is an example of estimating a one-dimensional functional (the ATE) of a complex Bayesian model (the full response surface). In such situations, the induced marginal posterior for the functional can often contain a significant bias in its centering, leading to poor estimation and uncertainty quantification [7, 8, 27]. This is indeed the case in our setting, where it is known that a naive choice of prior can yield badly biased inference for the ATE in casual inference/missing data problems [14, 26, 30]. For instance, Gaussian process (GP) priors will typically not be correctly centered, see Figure 1 below. Correcting for this is a delicate issue since even when the prior is perfectly calibrated (i.e. all tuning parameters are set optimally to recover the treatment response surface), the posterior can still induce a large bias in the marginal posterior for the ATE [25].

Our main contribution is to propose a data-driven modification to an arbitrary nonparametric prior based on the estimated propensity score that corrects for the first-order posterior bias for the ATE. By correctly centering the posterior for the ATE, this improves performance for both estimation accuracy and uncertainty quantification. We numerically illustrate our method on simulated and semi-synthetic data using GP priors, where our prior correction corresponds to a simple data-driven alteration to the covariance kernel. Our experiments demonstrate significant improvement in performance from this debiasing. This method should be viewed as a way to increase the efficiency of a given Bayesian prior, selected for modelling or computational reasons, when estimating the ATE.

Our method provides the same benefits for inference on the conditional average treatment effect (CATE). We further show that randomization of the feature distribution is not necessary for accurate uncertainty quantification for the CATE, but is helpful for the ATE. Since this approach provides similar estimation accuracy irrespective of whether the feature distribution is randomized, this highlights that care must be taken when using finer properties of the posterior, such as uncertainty quantification.

*Organization*: in Section 2 we present the causal inference problem, in Section 3 our main idea for debiasing an arbitrary Bayesian prior, with the specific case of GPs treated in Section 4. Simulations and further discussion are in Sections 5 and 6, respectively. Additional technical details, some motivation based on semiparametric statistics and further simulation results are in the supplement.

## 2   Problem setup

Consider the situation where a binary treatment with heterogeneous treatment effects is applied to a population. Working in the *potential outcomes* setup [19], every individual $i$ possesses a $d$-dimensional feature $X_i \in \mathbb{R}^d$ and two 'potential outcomes' $Y_i^{(1)}$ and $Y_i^{(0)}$, corresponding to the individual's outcomes with and without treatment, respectively. We wish to make inference on the treatment effect $Y_i^{(1)} - Y_i^{(0)}$, but since we only observe one out of each pair of outcomes, and not the corresponding (missing) *counterfactual* outcome, we do not directly observe samples of the treatment effect. In this paper we are interested in estimating the *average treatment effect* (ATE) $\psi = E[Y^{(1)} - Y^{(0)}]$.

For $R_i \in \{0, 1\}$ the treatment assignment indicator, we observe outcome $Y_i^{(R_i)}$, which can also be expressed as observing $Y = R_i Y^{(1)} + (1 - R_i) Y^{(0)}$. The treatment assignment policy generally depends on the features $X_i$ and is expressed by the conditional probability $\pi(x) = P(R = 1 | X = x)$ called the *propensity score* (PS). We assume *unconfoundedness*, namely $Y_i^{(1)}, Y_i^{(0)} \perp\!\!\!\perp R_i | X_i$ for all $X_i \in \mathbb{R}^d$, which is a standard assumption in the potential outcomes framework [19]. Unconfoundedness (or *strong ignorability*) says that the outcomes $Y_i^{(1)}, Y_i^{(0)}$ are independent of the treatment

assignment $R_i$ given the measured features $X_i$, i.e. any dependence can be fully explained through $X_i$. Without such an assumption the ATE is typically not even identifiable [32].

We work in the standard nonparametric regression framework for causal inference with mean-zero additive errors [2, 14, 15, 18, 23]

$$Y_i = m(X_i, R_i) + \varepsilon_i, \tag{1}$$

where $\varepsilon_i \sim^{iid} N(0, \sigma_n^2)$, $R_i \in \{0, 1\}$ is the indicator variable for whether treatment is applied and $X_i \in \mathbb{R}^d$ represents measured feature information about individual $i$. We assume the general feature information is unbiased $X_i \sim^{iid} F$, but the treatment assignment $\pi(x) = P(R = 1 | X = x)$ may be heavily biased. Our goal is to estimate the *average treatment effect* (ATE)

$$\psi = E[Y^{(1)} - Y^{(0)}] = \int_{\mathbb{R}^d} E[Y|R = 1, X = x] - E[Y|R = 0, X = x]dF(x)$$
$$= \int_{\mathbb{R}^d} m(x, 1) - m(x, 0)dF(x) \tag{2}$$

based on an observational dataset $\mathcal{D}_n$ consisting of $n$ i.i.d. samples of the triplet $(X_i, R_i, Y_i)$. A related quantity is the *conditional average treatment effect* (CATE)

$$\psi_c = \psi_c(X_1, \dots, X_n) = \frac{1}{n} \sum_{i=1}^{n} E[Y_i^{(1)} - Y_i^{(0)} | X_i] = \frac{1}{n} \sum_{i=1}^{n} m(X_i, 1) - m(X_i, 0), \tag{3}$$

which represents the average treatment effect over the measured individuals. Compared to the ATE, this quantity ignores the randomness in the feature data, replacing the true population feature distribution $F$ in the definition (2) of $\psi$ with its empirical counterpart $n^{-1} \sum_{i=1}^{n} \delta_{X_i}$ with $\delta_x$ the Dirac measure (point mass) at $x$.

## 3 Bayesian causal inference for average treatment effects

We fit a nonparametric prior to the model $(F, \pi, m)$ and consider the ATE $\psi$ as a functional of these three components, studying the one-dimensional marginal posterior for $\psi$ induced by the full nonparametric posterior. More concretely, one can sample from the marginal posterior for $\psi$ by drawing a full posterior sample $(F, \pi, m)$ and computing the corresponding draw $\psi$ according to the formula (2). Note that this yields the full posterior for the ATE $\psi$, which is much more informative than simply the posterior mean, for instance also providing credible intervals for $\psi$. This is the natural Bayesian approach to modelling $\psi$ and it is indeed typically necessary to fully model $(F, \pi, m)$ rather than $\psi$ directly when considering heterogeneous treatment effects.

Assuming the distribution $F$ has a density $f$, the likelihood for data $\mathcal{D}_n$ arising from model (1) is

$$\prod_{i=1}^{n} f(X_i)\pi(X_i)^{R_i}(1 - \pi(X_i))^{1-R_i} \frac{1}{\sqrt{2\pi}\sigma_n} e^{-\frac{1}{2\sigma_n^2} R_i(Y_i - m(X_i, 1))^2 - \frac{1}{2\sigma_n^2}(1-R_i)(Y_i - m(X_i, 0))^2}.$$

Since this factorizes in the model parameters $(f, \pi, m)$, placing a product prior on these three parameters yields a product posterior, i.e. $f$, $\pi$, $m$ are (conditionally) independent under the posterior. As this is particularly computationally efficient, we pursue this approach. In this case, since $\pi$ does not appear in the ATE $\psi$, the $\pi$ terms will cancel from the marginal posterior for $\psi$ and the prior on $\pi$ is irrelevant for estimating the ATE. We thus need not specify the $\pi$ component of the prior. These properties hold even when $F$ has no density and so a likelihood cannot be defined, see the supplement.

A Bayesian will typically endow the response surface $m$ with a nonparametric prior for either modelling or computational reasons. As already mentioned, the induced marginal posterior for $\psi$ will then often have a significant bias term in its centering, see Figure 1 for an example arising from a standard GP prior. Our main idea is to augment a given Bayesian prior for $m$ by efficiently using an estimate $\hat{\pi}$ of the PS, since it is well-known that using PS information can improve estimation of the ATE [32]. We model $(F, m)$ using the following prior:

$$m(x, r) = W(x, r) + \nu_n \lambda \left( \frac{r}{\hat{\pi}(x)} - \frac{1-r}{1 - \hat{\pi}(x)} \right), \qquad F \sim DP, \tag{4}$$

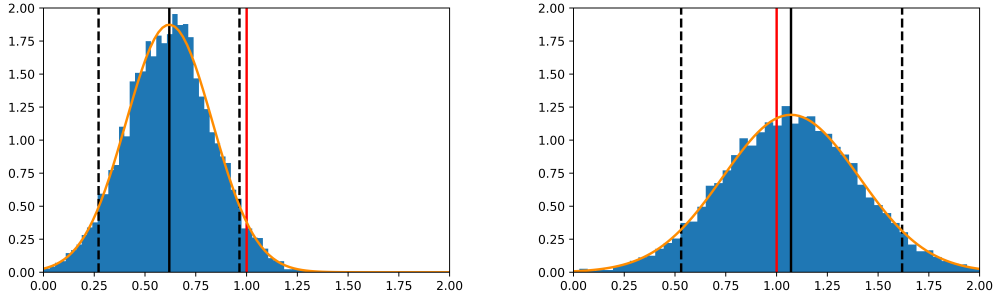

Figure 1: Plot of marginal posterior distributions for the ATE with true ATE (red), histogram of 10,000 posterior draws (blue), posterior mean (solid black), 90% credible interval (dotted black) and best fitting Gaussian distribution (orange). Data arises from the synthetic simulation (HOM) in Section 5 with $n = 500$ and Gaussian process prior described in Section 4. Left/right: without/with bias correction. Note the incorrect centering on the left-hand side.

where $W : \mathbb{R}^d \times \{0,1\} \to \mathbb{R}$ is a stochastic process, $DP$ denotes the Dirichlet process with a finite base measure [12], $\nu_n > 0$ is a scaling parameter and $\lambda$ is a real-valued random variable, with $W, F, \lambda$ independent. Estimating the PS is a standard binary classification problem and one can use any suitable estimator $\hat{\pi}$, from logistic regression to more advanced machine learning methods. It may be practically advantageous to truncate the estimator $\hat{\pi}$ away from 0 and 1 for numerical stability. For estimating the CATE, we propose the same prior (4) but with the Dirichlet process prior for $F$ replaced by a plug-in estimate consisting of the empirical distribution $F_n = n^{-1} \sum_{i=1}^n \delta_{X_i}$.

The prior (4) increases/decreases the prior correlation within/across treatment groups in a heterogeneous manner compared to the unmodified prior ($\nu_n = 0$). For example, in regions with few observations in the treatment group (small $\pi(x)$), (4) significantly increases the prior correlation with other treated individuals, thereby borrowing more information across individuals to account for the lack of data. Conversely, in observation rich areas (large $\pi(x)$), (4) borrows less information, instead using the (relatively) numerous local observations.

Using an unmodified prior ($\nu_n = 0$), the posterior will make a bias-variance tradeoff aimed at estimating the full regression function $m$ rather than the smooth one-dimensional functional $\psi$. In particular, the bias for the ATE $\psi$ will dominate, leading to poor estimation and uncertainty quantification unless the true $m$ and $f = F'$ are especially easy to estimate. The idea behind the prior (4) is to use a data-driven correction to (first-order) debias the resulting marginal posterior for $\psi$. The quantity $r/\pi(x) - (1 - r)/(1 - \pi(x))$ corresponds in a specific technical sense to the 'derivative' of the ATE $\psi$ with respect to the model (1), the so-called 'least favorable direction' of $\psi$. Heuristically, Taylor expanding $\psi|\mathcal{D}_n - \psi_0$, where $\psi|\mathcal{D}_n$ and $\psi_0$ are the posterior and 'true' ATE, the hyperparameter $\lambda$ is introduced to help the posterior remove the first-order (bias) term in this expansion, see Figure 1 for an illustration. Since the true $\pi$ is unknown, the natural approach is to replace it with an estimator $\hat{\pi}$. A more technical explanation can be found in the supplement.

Such a bias correction will help most when $(F, m)$ are difficult to estimate, for instance in high-dimensional feature settings. Higher-order bias corrections have also been considered using estimating equations [28, 29], but it is unclear how to extend this to the Bayesian setting. A similar idea has been investigated theoretically in [25], where it is shown that in a related idealized model, priors correctly calibrated to the unknown true functions (i.e. non-adaptive) satisfy a semiparametric Bernstein-von Mises theorem, i.e. the marginal posterior for the ATE is asymptotically normal with optimal variance in the large data setting. Figure 1 suggests the shape also holds in the present setting.

A good choice of prior for $W$ is still essential, since poor modelling of $m$ can also induce bias. In particular, $\nu_n$ should be picked so that the second term in (4) is of smaller order than $W$ in order to have relatively little effect on the full posterior for $m$. If the Bernstein-von Mises theorem holds, the marginal posterior for $\psi$ fluctuates on a $1/\sqrt{n}$ scale (see [25] for a related model), which suggests taking $\nu_n \sim 1/\sqrt{n}$. On this scale the bias correction is sufficiently large to meaningfully

affect the marginal posterior, but not so large as to dominate. Simulations indicate that taking $\nu_n$ significantly larger than this can cause the bias correction to dominate in small data situations, reducing performance. In a data-rich situation, larger values of $\nu_n$ are also admissible since the posterior can calibrate the value of $\lambda$ based on the data. Thus correct calibration of $\nu_n$ is mainly important for small or moderate sample performance, see Section 4.

One can also take a fully Bayesian approach by placing a prior on $\pi$ in (4). While such an approach may be philosophically appealing, it can cause computational difficulties since the priors for $(\pi, m)$, and hence also the corresponding posteriors, are no longer independent. For Gaussian processes (GPs), considered in detail in Section 4, one can then only sample from the fully Bayesian posterior using a Metropolis-Hastings-within-Gibbs-sampling algorithm, which is far slower in practice. In contrast, the 'empirical Bayes' approach we advocate in (4) maintains this independence and is thus computationally more efficient, e.g. in the GP case, the resulting prior for $m$ remains a GP.

It is known that for estimating a smooth one-dimensional functional of a nonparametric model, selecting an undersmoothing prior can be advantageous [8]. As well as being computationally efficient due to conjugacy, the choice of Dirichlet process for $F$ is thus also theoretically motivated, since it can be viewed as a considerable undersmoothing ($f = F'$ does not even exist as $F$ is a discrete probability measure with prior probability one).

One can also directly plug-in an estimator $F_n$ of $F$ in (2), such as the empirical distribution, and randomize only $m$ from its posterior. This provides an estimate of both the ATE (2) and CATE (3), but is only suitable for uncertainty quantification regarding the CATE. Not randomizing $F$ causes the posterior to ignore the uncertainty in the features, leading to an underestimation of the variance for $\psi$. The resulting credible intervals will then be too narrow, giving wrong uncertainty quantification as we see in the supplementary material. The message here is that even when different (empirical) Bayes methods give equally good estimation, as these two do, one must be careful about assuming that finer aspects of the posteriors behave similarly well, for example uncertainty quantification.

In summary, we view the prior modification (4) as a way to increase the efficiency of a given Bayesian prior for estimating the ATE and CATE.

A related approach is Bayesian Causal Forests (BCF) [15], where the estimated PS is directly added as an additional input feature to a BART model, yielding better performance. This approach is designed to improve nonparametric estimation of the *entire* response surface (i.e. the heterogeneous treatment effects themselves), which will also lead to some improvement when estimating the ATE. However, it is known that even when the prior is perfectly calibrated (i.e. all tuning parameters are set optimally) and recovers the entire response surface at the optimal rate, the posterior can still induce a bias in the *marginal posterior* for the ATE $\psi$ that prevents efficient estimation and destroys uncertainty quantification (see e.g. [25]).

As discussed above, the specific form in which we include the PS in our prior (4) is very deliberate, being motivated by semiparametric statistical theory and specifically designed for estimating the ATE. When either the PS or response surface are especially difficult to estimate, we expect that incorporating the PS as a feature as in BCF will still induce a bias for the ATE (the theory in [25] predicts this). We emphasize, however, that the main goal of BCF is to estimate the *entire* response surface, which is a different problem to estimating the ATE we consider here. An alternative Bayesian approach to estimating the ATE is to reparametrize the model to force $\pi$ into the likelihood [14, 26].

## 4 Gaussian process priors

In recent years, Gaussian process (GP) priors have found especial uptake in causal inference problems [1–4, 23], for example in healthcare [11, 38]. We therefore concretely illustrate the prior (4) for $W$ a mean-zero GP with covariance kernel $K$, $\lambda \sim N(0, 1)$ independent and scaling parameter $\nu_n > 0$ to be defined below. Under the prior (4), $m$ is again a mean-zero GP with data-driven covariance kernel

$$Em(x,r)m(x',r') = K((x,r),(x',r')) + \nu_n^2 \left( \frac{r}{\hat{\pi}(x)} - \frac{1-r}{1-\hat{\pi}(x)} \right) \left( \frac{r'}{\hat{\pi}(x')} - \frac{1-r'}{1-\hat{\pi}(x')} \right). \quad (5)$$

For GPs, our debiasing corresponds to a simple and easy to implement modification to the covariance kernel. One should use the original covariance kernel $K$ that was considered suitable for estimating $m$ (e.g. squared exponential, Matérn), since accurately modelling the regression surface is also necessary.

For our simulations in Section 5, we compute $\hat{\pi}$ using logistic regression based on the same data, truncating our estimator to $[0.1, 0.9]$ for numerical stability in (5). We take $K$ equal to the *squared exponential* kernel (also called *radial basis function*) with automatic relevance determination (ARD),

$$K((x, r), (x'r')) = \rho_m^2 \exp\left(-\frac{1}{2} \sum_{i=1}^{d} \frac{(x_i - x_i')^2}{\ell_i^2}\right) \exp\left(-\frac{1}{2} \frac{(r - r')^2}{\ell_{d+1}^2}\right)$$

with $(\ell_i)_{i=1}^{d+1}$ the length scale parameters and $\rho_m^2 > 0$ the kernel variance [24]. The data-driven length scales $\ell_i$ can be interpreted as the relevance of the $i^{th}$ feature to the regression surface $m$ and are particularly important for high-dimensional data, where some features may play little role. ARD has been used successfully for removing irrelevant inputs by several authors (see Chapter 5.1 [24]) and can thus be viewed as a form of automatic (causal) feature selection. We optimize the hyperparameters $(\ell_i)_{i=1}^{d+1}$, $\rho_m$ and $\sigma_n$ (noise variance) by maximizing the marginal likelihood (using the scaled conjugate gradient method option in the GPy package). We set $\nu_n = 0.2\rho_m/(\sqrt{n}M_n)$ for $M_n = n^{-1} \sum_{i=1}^{n} [R_i/\hat{\pi}(X_i) + (1 - R_i)/(1 - \hat{\pi}(X_i))]$ the average absolute value of the last part of (5). This places the second term in (5) on the same scale as the original covariance kernel $K$.

We assign $F|\mathcal{D}_n$ the Bayesian bootstrap (BB) distribution [12], namely a Dirichlet process with base measure equal to the rescaled empirical measure $\sum_{i=1}^{n} \delta_{X_i}$ of the observations. When $n$ is moderate or large, the BB distribution will be very close to that of the true DP posterior. The advantage of the BB is that samples are particularly easy to generate: using that $F|\mathcal{D}_n$ can be represented as $\sum_{i=1}^{n} V_i \delta_{X_i}$ for $(V_1, \ldots, V_n) \sim \text{Dir}(n; 1, \ldots, 1)$ and that $m$ and $F$ are independent under the posterior, the posterior mean and draws for the ATE can be written as

$$E[\psi|\mathcal{D}_n] = \frac{1}{n} \sum_{i=1}^{n} E\left[m(X_i, 1) - m(X_i, 0)|\mathcal{D}_n\right], \quad \psi|\mathcal{D}_n = \sum_{i=1}^{n} V_i \left(m(X_i, 1) - m(X_i, 0)\right), \quad (6)$$

respectively. Using the representation $V_i = U_i/\sum_{j=1}^{n} U_j$ for $U_i \sim^{iid} \exp(1)$, sampling $(V_1, \ldots, V_n)$ is particularly simple. One also needs to generate an $n$-dimensional multivariate Gaussian random variable $(m(X_i, 1) - m(X_i, 0))_{i=1}^{n}$, whose covariance can be directly obtained from the posterior GP process $(m(x, r) : x \in \mathbb{R}^d, r \in \{0, 1\})|\mathcal{D}_n$ evaluated at the observations and their counterfactual values. This follows from the usual formula for the mean and covariance of a posterior GP in regression with Gaussian noise (Chapter 2.2 of [24]) and the whole procedure is summarized in Algorithm 1[1]. Using this scheme, we may sample directly from the marginal posterior for the ATE $\psi$.

To show the importance of randomizing $F$ for uncertainty quantification, we also consider the posterior where one plugs in the empirical measure $n^{-1} \sum_{i=1}^{n} \delta_{X_i}$ for $F$ in (2). This yields the same posterior mean as in (6), while sampling $\psi|\mathcal{D}_n$ corresponds to the right-hand side of (6) with $V_i$ replaced by $1/n$. We expect this to yield similar prediction to the posterior mean in (6) but worse uncertainty quantification for the ATE (but not CATE). This is indeed what we see in the supplement.

## 5   Simulations

We numerically illustrate the improved performance of our debiased GP method (GP+PS) versus the original GP approach, both with (GP and GP+PS) and without randomization (GP (noRand) and GP + PS (noRand)) of the feature distribution $F$. The methods are implemented as described in Section 4. Credible intervals are computed by sampling 2,000 posterior draws and taking the empirical 95% credible interval, see Figure 1. We measure estimation accuracy via the absolute error between the posterior mean and true (C)ATE. We also report the average size and coverage of the resulting credible/confidence intervals (CI) and the Type II error, which measures the fraction of times the method does not identity a statistically significant (C)ATE.

We further compare their performance with standard state-of-art-methods for estimating the ATE and CATE, namely Bayesian Additive Regression Trees (BART) [10, 18, 15] both with and without using the PS as a feature, Bayesian Causal Forests (BCF) [15], Causal Forests (CF) with average inverse

**Algorithm 1** Debiased GP with PS correction

---

1: **Input:** $X$ (features), $R$ (treatment assignments), $Y$ (outcomes), $K$ (covariance kernel)
2: Run logistic regression on $(X_1, R_1), \ldots, (X_n, R_n)$ and return estimates $\hat{\pi}(X_1), \ldots, \hat{\pi}(X_n)$
3: $\mathbf{w_f} = \left( \frac{R_1}{\hat{\pi}(X_1)} - \frac{1-R_1}{1-\hat{\pi}(X_1)}, \ldots, \frac{R_n}{\hat{\pi}(X_n)} - \frac{1-R_n}{1-\hat{\pi}(X_n)} \right)$ (factual)
4: $\mathbf{w_c} = \left( \frac{1-R_1}{\hat{\pi}(X_1)} - \frac{R_1}{1-\hat{\pi}(X_1)}, \ldots, \frac{1-R_n}{\hat{\pi}(X_n)} - \frac{R_n}{1-\hat{\pi}(X_n)} \right)$ (counterfactual)
5: Optimize hyperparameters of $k$ (including $\sigma_n^2$) and then $\nu^2$ (see Section 4)
6: $Z = (X\ R)$ and $Z_* = \begin{pmatrix} X & R \\ X & 1-R \end{pmatrix}$
7: $\overline{K}_{f,c} = K(Z_*, Z) + \nu^2 (\mathbf{w_f}\ \mathbf{w_c})^T \mathbf{w_f}$
8: $\boldsymbol{\mu} = \overline{K}_{f,c}[K(Z,Z) + \nu^2 \mathbf{w_f}^T \mathbf{w_f} + \sigma_n^2 I_n]^{-1} Y$
9: $\boldsymbol{\Sigma} = K(Z_*, Z_*) + \nu^2 (\mathbf{w_f}\ \mathbf{w_c})^T (\mathbf{w_f}\ \mathbf{w_c}) - \overline{K}_{f,c}[K(Z,Z) + \nu^2 \mathbf{w_f}^T \mathbf{w_f} + \sigma_n^2 I_n]^{-1} \overline{K}_{f,c}^T$
10: Compute $\hat{\psi} = E[\psi|\mathcal{D}_n]$ from $\boldsymbol{\mu}$ according to the left hand side of (6)
11: **for** $l = 1 \ldots P$ (# posterior samples) **do**
12:      Generate $(V_1, \ldots, V_n) \sim \text{Dir}(n; 1, \ldots, 1)$
13:      Generate $\boldsymbol{m} \sim N_{2n}(\boldsymbol{\mu}, \boldsymbol{\Sigma})$
14:      Compute $\psi_l$ from $\boldsymbol{m}$ and $V_1, \ldots, V_n$ according to the right hand side of (6)
15: Compute credible interval (CI) based on quantiles of $\psi_1, \ldots, \psi_P$
16: **Output:** $\hat{\psi}$ (posterior mean), CI (credible interval), $\psi_1, \ldots, \psi_P$ (posterior samples)

---

propensity weighting (AIPW) and targeted maximum likelihood estimation (TMLE) [39], Propensity Score Matching (PSM) [36], ordinary least squares (OLS), and Covariate Balancing (CB) with the standard inverse PS weights and weights computed by constrained minimization (CM) [9]. Details of these benchmarks are provided in the supplementary material. We ran all simulations 200 times and report average values.

**Synthetic dataset.** We consider two versions of synthetic data generated following the protocol used in [21, 22, 37]. We take sample sizes $n = 500, 1000$ and $d = 100$ features $x_1, x_2, ..., x_{100} \overset{iid}{\sim} N(0,1)$. The response surface and treatment assignments are defined via the following ten functions: $g_1(x) = x - 0.5$, $g_2(x) = (x - 0.5)^2 + 2$, $g_3(x) = x^2 - 1/3$, $g_4(x) = -2\sin(2x)$, $g_5(x) = e^{-x} - e^{-1} - 1$, $g_6(x) = e^{-x}$, $g_7(x) = x^2$, $g_8(x) = x$, $g_9(x) = I_{x>0}$, $g_{10}(x) = \cos(x)$. A subject with features $x = (x_1, ..., x_{100})$ is assigned (non-randomly) to the treatment group if $\sum_{k=1}^{5} g_k(x_k) > 0$ and otherwise to the control group. Given the features and treatment assignment, in case (HOM) the outcome $Y$ is generated as $Y|X = x, R = r \sim N(\sum_{k=1}^{5} g_{k+5}(x_k) + r, 1)$, which models a homogeneous treatment effect. In case (HET), $Y$ is generated as $Y|X = x, R = r \sim N(\sum_{k=1}^{5} g_{k+5}(x_k) + r(1 + 2x_2 x_5), 1)$, which models heterogeneous treatment effects. In both cases, the first five features affect both the treatment and outcome, representing confounders, while the remaining 95 features are noise. The ATE is 1 in both cases. Some results are in Table 1 with the remainder in the supplement.

**IHDP dataset with simulated outcomes.** Since simulated covariates often do not accurately represent "real world" examples, we consider a semi-synthetic dataset with real features and treatment assignments from the Infant Health and Development Program (IHDP), but simulated responses. The IHDP consisted of a randomized experiment studying whether low-birth-weight and premature infants benefited from intensive high-quality child care. The data contains $d = 25$ pretreatment variables per subject. Following [18] (also used in [2, 21]), an observational study is created by removing a non-random portion of the treatment group, namely all children with non-white mothers. This leaves a dataset of 747 subjects, with 139 in the treatment group and 608 in the control group.

We consider a slight modification of the non-linear "Response Surface B" of [18], taking

$$Y^{(0)}|X = x \sim N(e^{(x+w)\beta}, 1) \quad \text{and} \quad Y^{(1)}|X = x \sim N(x^T\beta - \omega_\beta, 1),$$

where $x \in \mathbb{R}^d$ are the features, $w = (0.5, \ldots, 0.5)$ is an offset vector, $\beta$ is a vector of regression coefficients with each entry randomly sampled from $\{0, 0.1, 0.2, 0.3, 0.4\}$ with probabilities $(0.6, 0.1, 0.1, 0.1, 0.1)$. For each simulation of $\beta$, $\omega_\beta$ is then selected so that the CATE equals 4. Here, we can only measure estimation quality of the CATE and not the ATE since the true feature distribution $F$ is unknown. Results are in Table 2.

Table 1: Results for synthetic dataset (HET) with $n = 1000$.

| Method | Abs. error $\pm$ sd | Size CI $\pm$ sd | Coverage | Type II error |
|---|---|---|---|---|
| GP | $0.321 \pm 0.027$ | $0.613 \pm 0.027$ | 0.38 | **0.00** |
| GP (noRand) | $0.321 \pm 0.027$ | $0.427 \pm 0.017$ | 0.00 | **0.00** |
| GP PS | **$0.063 \pm 0.042$** | $0.883 \pm 0.040$ | **1.00** | **0.00** |
| GP PS (noRand) | **$0.063 \pm 0.042$** | $0.766 \pm 0.037$ | **1.00** | **0.00** |
| BART | $0.228 \pm 0.186$ | $1.723 \pm 0.490$ | **1.00** | 0.50 |
| BART (PS) | $0.134 \pm 0.092$ | $0.741 \pm 0.079$ | 0.99 | **0.00** |
| BCF | $0.144 \pm 0.109$ | $0.535 \pm 0.066$ | 0.87 | **0.00** |
| CF (AIPW) | $0.138 \pm 0.097$ | **$0.695 \pm 0.103$** | 0.96 | **0.00** |
| CF (TMLE) | $0.136 \pm 0.099$ | $0.891 \pm 0.156$ | 0.99 | 0.01 |
| OLS | $0.725 \pm 0.160$ | $0.361 \pm 0.034$ | 0.00 | 0.26 |
| CB (IPW) | $0.606 \pm 0.324$ | $1.467 \pm 0.418$ | 0.68 | 0.01 |
| PSM | $0.234 \pm 0.178$ | $1.282 \pm 0.158$ | 0.97 | 0.06 |

Table 2: Results for semi-synthetic IHDP dataset.

| Method | Abs. error $\pm$ sd | Size CI $\pm$ sd | Coverage | Type II error |
|---|---|---|---|---|
| GP | $0.246 \pm 0.398$ | $1.383 \pm 1.458$ | 0.95 | 0.01 |
| GP (noRand) | $0.246 \pm 0.398$ | $1.096 \pm 1.305$ | 0.89 | 0.01 |
| GP + PS | $0.189 \pm 0.234$ | $1.445 \pm 1.013$ | 0.97 | 0.01 |
| GP +PS (noRand) | $0.189 \pm 0.234$ | $1.162 \pm 0.822$ | 0.93 | 0.01 |
| BART | $0.234 \pm 0.282$ | $0.945 \pm 0.745$ | 0.91 | **0.00** |
| BART (PS) | $0.238 \pm 0.342$ | $0.906 \pm 0.682$ | 0.89 | **0.00** |
| BCF | **$0.108 \pm 0.106$** | **$0.526 \pm 0.151$** | 0.95 | **0.00** |
| CF (AIPW) | $0.245 \pm 0.236$ | $1.052 \pm 0.811$ | 0.91 | 0.01 |
| CF (TMLE) | $0.242 \pm 0.240$ | $1.087 \pm 0.842$ | 0.91 | 0.01 |
| OLS | $0.127 \pm 0.101$ | $0.815 \pm 0.537$ | 0.98 | **0.00** |
| CB (IPW) | $0.238 \pm 0.180$ | $1.200 \pm 0.860$ | 0.91 | **0.00** |
| CB (CM) | $0.134 \pm 0.117$ | $0.961 \pm 0.765$ | 0.93 | **0.00** |
| PSM | $0.136 \pm 0.108$ | $2.052 \pm 1.701$ | **1.00** | 0.01 |

Both of these simulations contain unbalanced treatment groups, with roughly 90% and 20% of subjects in the treatment group in the synthetic and IHDP simulations, respectively. Like PS reweighting-based methods, our bias corrected GP method (5) is designed with problems satisfying the standard *overlap* assumption [19] (namely $0 < P(R = 1|X = x) < 1$ for all $x \in \mathbb{R}^d$) in mind. In the synthetic simulation the treatment assignment is fully deterministic so this condition is not satisfied; in particular the data generation process was not selected to favour our method.

**Results.** We see from Tables 1 and 2 that our methods (GP + PS and GP + PS (noRand)) substantially improve upon the performance of the vanilla GP methods (GP and GP (noRand)) [also true in the additional simulations in the supplement]. In both cases we obtain significantly improved estimation accuracy and uncertainty quantification. As an example of what can go wrong, in the synthetic simulation the absolute errors of the vanilla GP methods barely decrease as the sample size increases (Table 1 and the supplementary tables) since the posterior for the ATE contains a non-vanishing bias. Moreover, since the posterior variance shrinks rapidly with the sample size (at rate $1/n$ for the ATE [25]), the posterior will concentrate tightly around the wrong value, giving poor uncertainty quantification that actually worsens with increasing data, see Figure 1 and Table 1. This is a typical aspect of causal inference problems with difficult to estimate PS and response surfaces, particularly

in high feature dimensions. In contrast, our debiased method explicitly corrects for this bias at the expense of a (smaller) increase in variance, as can be seen from the average CI length. The substantially improved coverage from our method is the result of the debiasing rather than the increase in posterior variance.

Asymptotic theory predicts the frequentist coverage of our method should converge to exactly 0.95 as the sample size increases due to the semiparametric Bernstein-von Mises theorem [25]. However, it is a subtle question as to when the asymptotic regime applies and our examples seem insufficiently data rich for this to be the case (e.g. $d = 100$ input features, but only $n = 1000$ observations).

We see that our method makes the previously underperforming GP method highly competitive with state-of-the-art methods, even outperforming them in certain cases. In the synthetic simulation, our method performs best yielding substantially better estimation accuracy. It further provides reliable and informative uncertainty quantification, performing similarly to BART (PS), CF (AIPW) and CF (TMLE). On the IHDP dataset, our debiased methods outperform the widely used BART and CF for estimation accuracy, but BCF performs best. While OLS and CB (CM) also performed well here, we note that in the synthetic simulation, OLS performed especially badly while CB (CM) did not even run. Regarding uncertainty quantification, our method provides excellent coverage though larger CIs than BART (whose coverage is slightly lower), but BCF again performs best.

We lastly note that not randomizing the feature distribution (noRand) yields narrower CIs and lower coverage as expected. This does not make a substantial difference in Tables 1 and 2, but can have a significant impact, see the tables in the supplement. We recall that randomization is generally helpful for uncertainty quantification for the ATE, but is conservative for the CATE.

## 6   Discussion

We have introduced a general data-driven modification that can be applied to any given prior that corrects for first-order posterior bias when estimating (conditional) average treatment effects (ATEs) in a causal inference regression model. We illustrated this experimentally on both simulated and semi-synthetic data for the example of Gaussian process (GP) priors. We showed that by correctly incorporating an estimate of the propensity score into the covariance kernel, one can substantially improve the precision of both the posterior mean and posterior uncertainty quantification. In particular, this makes the modified GP method highly competitive with state-of-the-art methods.

There are many avenues for future work. First, GP methods scale poorly with data size and there has been extensive research on scalable alternatives, including sparse GP approximations, variational Bayes and distributed computing approaches. Since in the GP case our approach simply returns a GP with modified covariance kernel, all these existing methods should be directly applicable and can be investigated. Second, it would be particularly interesting to see if our prior correction can be efficiently implemented to improve the already excellent performance of BART and its derivatives in causal inference problems [15, 18]. Third, it is unclear if and how one can perform higher order bias corrections using Bayes for especially difficult problems as has been done using estimating equations [28, 29].

**Acknowledgements**: Botond Szabó received funding from the Netherlands Organization for Scientific Research (NWO) under Project number: 639.031.654. We thank 3 reviewers for their useful comments that helped improve the presentation of this work.

## Footnotes

[1]Lines 8-9 in Algorithm 1 are the usual predictive mean and covariance computations for a posterior GP. In particular, these can be more efficiently solved using for example Cholesky factorization, see Chapter 2.2 of [24]. Similarly, $m$ can be efficiently generated by once taking the Cholesky factor $L_\Sigma$ of $\Sigma$, generating $W \sim N_{2n}(0, I_{2n})$ and setting $m = \mu + L_\Sigma W$

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
