[Supplementary Material]

# Supplementary Material for "Debiased Bayesian inference for average treatment effects"

**Kolyan Ray**
Department of Mathematics
King's College London
kolyan.ray@kcl.ac.uk

**Botond Szabó**
Mathematical Institute
Leiden University
b.t.szabo@math.leidenuniv.nl

## Appendix A: A product prior leads to a product posterior

Suppose that we place an independent prior on the three model parameters $F$, $\pi$ and $m$. Assuming the feature distribution $F$ has a density $f$, we recall that the likelihood for data $\mathcal{D}_n = \{(X_i, R_i, Y_i)\}_{i=1}^n$ arising from model (2) is

$$\prod_{i=1}^n f(X_i)\pi(X_i)^{R_i}(1-\pi(X_i))^{1-R_i}\frac{1}{\sqrt{2\pi}\sigma_n}e^{-\frac{1}{2\sigma_n^2}R_i(Y_i-m(X_i,1))^2 - \frac{1}{2\sigma_n^2}(1-R_i)(Y_i-m(X_i,0))^2}. \quad (1)$$

Since this factorizes as $L_n^f L_n^\pi L_n^m$, where each term is a function of only $f$, $\pi$ and $m$, respectively, the posterior will again factorize so that $f$, $\pi$ and $m$ are also independent under the posterior (i.e. conditional on the data). We assign $F$ a Dirichlet process prior, which does not give probability one to a dominated set of measures, meaning that the posterior of $(F, \pi, m)$ cannot be derived using Bayes formula. Nonetheless, we can still obtain the form of the posterior, in particular establishing the posterior independence of the three parameters.

The following argument is found in Section 3.1 of [4] and we reproduce it for the convenience of the reader. It is well-known that in the model consisting of sampling $F$ from the Dirichlet process prior with base measure $\alpha$ and next sampling observations $X_1, \ldots, X_n$ from $F$, the posterior of $F|X_1\ldots,X_n$ is again a Dirichlet process with updated base measure $\alpha + nP_n$, where $P_n = n^{-1}\sum_{i=1}^n \delta_{X_i}$ is the empirical distribution of $X_1, \ldots, X_n$ (see Chapter 4 of [3] for further details). Let $Q$ denote the prior distribution of $(\pi, m)$. The parameters $(F, \pi, m)$ and data are generated via the hierarchical scheme:

- $F \sim DP(\alpha)$ and $(\pi, m) \sim Q$ independently.
- The features satisfy $X_1, \ldots, X_n|(F, \pi, m) \sim^{iid} F$.
- $R_i|(F, \pi, m, X_1, \ldots, X_n) \sim \text{Bin}(1, \pi(X_i))$ and $Y_i^{(t)}|(F, \pi, m, X_1, \ldots, X_n) \sim N(m(X_i, t), \sigma_n^2)$, $t = 0, 1$, are (conditionally) independent.
- The observations are $\mathcal{D}_n = \{(X_1, R_1, Y_1), \ldots, (X_n, R_n, Y_n)\}$ with $Y_i = R_i Y_i^{(1)} + (1 - R_i)Y_i^{(0)}$.

Using this scheme, we can observe that $F$ and $(R_1, \ldots, R_n, Y_1, \ldots, Y_n)$ are conditionally independent given $(\pi, m, X_1, \ldots, X_n)$. Similarly, $F$ and $(\pi, m)$ are conditionally independent given $\mathcal{D}_n$. It thus follows that the posterior distribution of $F$ given $\mathcal{D}_n$ is identical to the posterior of $F$ given $X_1, \ldots, X_n$, namely the $DP(\alpha + nP_n)$ distribution. Moreover, the posterior of $(\pi, m)$ given $(F, \mathcal{D}_n)$ can then be obtained using Bayes rule with the likelihood of $(R_1, \ldots, R_n, Y_1, \ldots, Y_n)$ given $X_1, \ldots, X_n$, which is indeed dominated. Denoting the full prior by $\Pi$, this yields posterior distribution

$$\Pi\left((\pi, m) \in A, F \in B|\mathcal{D}_n\right) = \int_B \frac{\int_A L_n^\pi L_n^m d\Pi(\pi, m)}{\int L_n^\pi L_n^m d\Pi(\pi, m)}d\Pi(F|X_1, \ldots, X_n),$$

where $L_n^\pi L_n^m$ is the likelihood (1) with $f$ set to 1. Since the above integrals separate, the three parameters are independent under the posterior (since also $\pi$ and $m$ were assumed independent under the prior). The above formula also extends to the Bayesian bootstrap (BB), which has base measure $\alpha = 0$, which we consider in our simulations.

## Appendix B: A technical motivation for the prior correction

We provide a brief technical motivation for readers familiar with semiparametric estimation theory (see for example Chapter 25 of [7]). Recall that we work in the nonparametric regression model:

$$Y_i = m(X_i, R_i) + \varepsilon_i, \tag{2}$$

where $\varepsilon_i \sim^{iid} N(0, \sigma_n^2)$, $R_i \in \{0, 1\}$ and $X_i \in \mathbb{R}^d$ represents measured feature information about individual $i$. We further assume that $X_i \sim^{iid} F$ and define the propensity score $\pi(x) = P(R = 1|X = x)$. Our goal is to estimate the ATE $\psi = \int_{\mathbb{R}^d} m(x, 1) - m(x, 0) dF(x)$.

We very briefly summarize some facts concerning the semiparametric estimation theory of the ATE $\psi$ in the model (2) (see e.g. [5]). Let $\Psi(t) = 1/(1 + e^{-t})$ denote the logistic function. Consider the one-dimensional submodels $t \mapsto (f_t, \pi_t, m_t)$ of (2) defined via the paths

$$f_t = f e^{t\phi - \log \int f e^{t\phi}}, \quad \pi_t = \Psi(\Psi^{-1}(\pi) + t\alpha), \quad m_t(x, r) = m(x, r) + t\gamma(x, r),$$

for given 'directions' (functions) $(\phi, \alpha, \gamma)$ with $\int \phi f = 0$. Set $q_t = (f_t, \pi_t, m_t)$. The difficulty of estimating $\psi(q)$ in the one-dimensional submodel $\{q_t : t \in (-\varepsilon, \varepsilon)\}$ depends on the functions $(\phi, \alpha, \gamma)$. This can be quantified via the best possible asymptotic variance achievable by any estimator when estimating $\psi(q)$ in this model, with larger such variance indicating a more difficult problem. The most difficult such submodel, if it exists, has the largest asymptotically optimal variance for estimating $\psi$ and the corresponding functions $(\phi, \alpha, \gamma)$ are called the 'least favourable direction'. In model (2) this equals

$$(\phi, \alpha, \gamma) = \left( m(X, 1) - m(X, 0) - \psi(q), 0, \frac{R\sigma_n^2}{\pi(X)} - \frac{(1 - R)\sigma_n^2}{1 - \pi(X)} \right),$$

where one can see the third term mirrors our prior correction.

It is known from the Bayesian nonparametric asymptotics literature that a condition for the semiparametric Bernstein-von Mises theorem (Chapter 12 of [3]) to hold, and hence for the marginal posterior for $\psi$ to be statistically optimal in a frequentist sense, is that the prior be invariant under a shift of the nonparametric component in the least favourable direction [1, 2]. The prior correction for our method, which takes the form of the (estimated) least favourable direction, exactly provides such an invariance by giving the prior an explicit component in this direction (otherwise the shift may be in some sense 'orthogonal' to the underlying prior). One can thus view the bias correction as an attempt to provide additional robustness against posterior inaccuracy in the 'most difficult direction', namely the one which will induce the largest bias in the ATE $\psi$. For a theoretical analysis of such an idea in a related idealized model, see [4].

## Appendix C: Brief description of the benchmark methods

The BART method consists of two parts: a sum-of-trees model on the response surface and a prior distribution on the trees for regularization. One can use MCMC methods to compute summary statistics (e.g. point estimators, credible sets), which in practice provides a stable solution. For implementation we use the "bartCause" R package and consider two alternatives. First, we fit a BART model to both the treatment variable and response surface (we call the bartc() function with arguments method.rsp="bart" and method.trt="bart"). Second, we fit a BART to the response surface with the propensity score (estimated with logistic regression) included as a predictor, and use propensity score weighted averages of the treatment effect to estimate the ATE (we call the bartc() function with arguments method.rsp="p.weight" and method.trt="glm").

In Bayesian Causal Forests (BCF) the estimated PS is added as an additional input feature to a BART model. In the implementation we use the "bcf" R package to estimate the ATE and call the bcf() function with $nsim = 2000$ and $nburn = 2000$. The propensity score is estimated via logistic regression using the $glm()$ function.

Table 1: Results for synthetic dataset (HET) with $n = 500$.

| Method | Abs. error ± sd | Size CI ± sd | Coverage | Type II error |
|---|---|---|---|---|
| GP | $0.319 \pm 0.042$ | $\mathbf{0.871 \pm 0.043}$ | 0.98 | **0.0** |
| GP (noRand) | $0.319 \pm 0.042$ | $0.606 \pm 0.029$ | 0.36 | **0.0** |
| GP PS | $\mathbf{0.106 \pm 0.081}$ | $1.368 \pm 0.090$ | **1.00** | **0.00** |
| GP PS (noRand) | $\mathbf{0.106 \pm 0.081}$ | $1.218 \pm 0.086$ | **1.00** | **0.00** |
| BART | $0.712 \pm 0.371$ | $2.251 \pm 0.571$ | 0.98 | 0.91 |
| BART (PS) | $0.239 \pm 0.185$ | $1.225 \pm 0.157$ | 0.96 | 0.24 |
| BCF | $0.234 \pm 0.182$ | $0.830 \pm 0.127$ | 0.83 | 0.06 |
| CF (AIPW) | $0.181 \pm 0.139$ | $1.059 \pm 0.190$ | 0.98 | 0.07 |
| CF (TMLE) | $0.183 \pm 0.137$ | $1.223 \pm 0.246$ | 0.99 | 0.08 |
| OLS | $0.460 \pm 0.261$ | $0.855 \pm 0.112$ | 0.48 | 0.30 |
| CB (IPW) | $0.422 \pm 0.314$ | $1.726 \pm 0.421$ | 0.93 | 0.09 |
| PSM | $0.295 \pm 0.225$ | $1.729 \pm 0.232$ | 0.98 | 0.29 |

In the Causal Forest methods, we train a random forest to estimate the response surface and then estimate the ATE by using either average inverse propensity weighting (AIPW) or targeted maximum likelihood estimation (TMLE). In the implementation we use the "grf" R package and call the average_treatment_effect() function with arguments model="AIPW" and model="TMLE", respectively.

In the propensity score matching algorithm, the PS $\pi(x)$ is first estimated, for instance by logistic regression, and then the samples in the treatment and control groups are matched based on the estimated PS. The key idea behind this approach is that "if a subclass of units or a matched treatment-control pair is homogeneous in $\pi(x)$, then the treated and control units in that subclass or matched pair will have the same distribution of $x$" [6], or in other words the treatment and control groups will be balanced in the covariates. Then the average treatment effect can be easily estimated using the matched pairs even in case of large dimensional feature spaces. To obtain balanced covariates, one typically has to use an iterative algorithm while assessing at each iteration the balance of the features in the control and treatment groups and correcting the propensity score estimates accordingly. In the implementation we use the "Matching" R package and call the Match() function with arguments estimand= "ATE", $Z = x$ and $M = 1$.

Ordinarily Least Squares estimator for ATE is the difference between the predicted value of the linear least squares estimators in the treatment and control groups. This is a simple, straightforward method which works well only for models close to linear.

In our analysis we consider two type of CB methods, one based on the standard inverse propensity score weighting and the other on the constrained optimization method. For the former one we used the "balanceHD" R package calling the ipw.ate() with arguments prop.method = "elnet", fit.method = "none", prop.weighted.fit = T and targeting.method = "AIPW". For the second one we have applied the "ATE" R package and called the ATE() function.

## Appendix D: Additional simulation results

We provide the remaining numerical results for the synthetic simulations with heterogeneous treatment effects ($n = 500$) and homogeneous treatment effects ($n = 500, 1000$). Note that for $n = 500$ (Tables 1 and 2) not randomizing the feature distribution $F$ (noRand) leads to a dramatic drop in coverage when using a vanilla GP to perform uncertainty quantification for the ATE.

## References

[1] CASTILLO, I. A semiparametric Bernstein–von Mises theorem for Gaussian process priors. *Probab. Theory Related Fields 152*, 1-2 (2012), 53–99.

Table 2: Results for synthetic dataset (HOM) with $n = 500$.

| Method | Abs. error $\pm$ sd | Size CI $\pm$ sd | Coverage | Type II error |
|---|---|---|---|---|
| GP | $0.314 \pm 0.036$ | $0.826 \pm 0.048$ | 0.98 | **0.00** |
| GP (noRand) | $0.314 \pm 0.036$ | $0.575 \pm 0.034$ | 0.29 | **0.00** |
| GP PS | **$0.119 \pm 0.095$** | $1.305 \pm 0.090$ | **1.00** | **0.00** |
| GP PS (noRand) | **$0.119 \pm 0.095$** | $1.161 \pm 0.088$ | **1.00** | **0.00** |
| BART | $0.239 \pm 0.186$ | $1.760 \pm 0.558$ | 0.99 | 0.52 |
| BART (PS) | $0.123 \pm 0.095$ | $0.684 \pm 0.061$ | 0.97 | **0.00** |
| BCF | $0.195 \pm 0.298$ | **$0.657 \pm 0.140$** | 0.840 | 0.02 |
| CF (AIPW) | $0.187 \pm 0.148$ | $0.901 \pm 0.161$ | 0.92 | 0.01 |
| CF (TMLE) | $0.185 \pm 0.147$ | $1.057 \pm 0.210$ | 0.95 | 0.03 |
| OLS | $0.437 \pm 0.236$ | $0.817 \pm 0.101$ | 0.45 | 0.25 |
| CB (IPW) | $0.489 \pm 0.274$ | $1.434 \pm 0.317$ | 0.74 | 0.03 |
| PSM | $0.312 \pm 0.227$ | $1.477 \pm 0.204$ | 0.94 | 0.20 |

Table 3: Results for synthetic dataset (HOM) with $n = 1000$.

| Method | Abs. error $\pm$ sd | Size CI $\pm$ sd | Coverage | Type II error |
|---|---|---|---|---|
| GP | $0.312 \pm 0.025$ | $0.584 \pm 0.022$ | 0.23 | **0.00** |
| GP (noRand) | $0.312 \pm 0.025$ | $0.406 \pm 0.016$ | 0.00 | **0.00** |
| GP PS | **$0.057 \pm 0.042$** | $0.841 \pm 0.038$ | **1.00** | **0.00** |
| GP PS (noRand) | **$0.057 \pm 0.042$** | $0.726 \pm 0.035$ | **1.00** | **0.00** |
| BART | $0.150 \pm 0.143$ | $1.172 \pm 0.414$ | **1.00** | 0.17 |
| BART (PS) | $0.072 \pm 0.045$ | $0.445 \pm 0.027$ | 0.96 | **0.00** |
| BCF | $0.098 \pm 0.086$ | **$0.431 \pm 0.155$** | 0.89 | 0.01 |
| CF (AIPW) | $0.145 \pm 0.107$ | $0.595 \pm 0.114$ | 0.85 | **0.00** |
| CF (TMLE) | $0.141 \pm 0.142$ | $0.791 \pm 0.170$ | 0.96 | **0.00** |
| OLS | $0.706 \pm 0.156$ | $0.351 \pm 0.028$ | 0.00 | 0.22 |
| CB (IPW) | $0.634 \pm 0.262$ | $1.171 \pm 0.421$ | 0.39 | **0.00** |
| PSM | $0.238 \pm 0.179$ | $1.104 \pm 0.114$ | 0.92 | 0.01 |

[2] CASTILLO, I., AND ROUSSEAU, J. A Bernstein–von Mises theorem for smooth functionals in semiparametric models. *Ann. Statist. 43*, 6 (2015), 2353–2383.

[3] GHOSAL, S., AND VAN DER VAART, A. W. *Fundamentals of Nonparametric Bayesian Inference.* Cambridge Series in Statistical and Probabilistic Mathematics. Cambridge University Press, Cambridge, 2017.

[4] RAY, K., AND VAN DER VAART, A. W. Semiparametric Bayesian causal inference. *Ann. Statist.,* to appear.

[5] ROBINS, J. M., LI, L., MUKHERJEE, R., TCHETGEN, E. T., AND VAN DER VAART, A. Minimax estimation of a functional on a structured high-dimensional model. *Ann. Statist. 45*, 5 (2017), 1951–1987.

[6] ROSENBAUM, P. R., AND RUBIN, D. B. The central role of the propensity score in observational studies for causal effects. *Biometrika 70*, 1 (1983), 41–55.

[7] VAN DER VAART, A. W. *Asymptotic statistics*, vol. 3 of *Cambridge Series in Statistical and Probabilistic Mathematics.* Cambridge University Press, Cambridge, 1998.