[Reviews · NeurIPS 2019]

Reviewer 1



The paper presents a Bayesian framework for causal inference in observational studies, in the potential outcomes setup. When studying the average treatment effects, results based on Gaussian process priors may be biased and the paper proposes a correction. I have the impression that the paper does not pursue a fully Bayesian approach since \pi does not seem to be estimated and its prior distribution is never defined. In particular, contrary to a standard Bayesian approach, it is separately estimated and its estimate is plugged-in the estimation procedure for m(x,r). A comparison with a fully Bayesian approach is missing. The bias the authors want to correct is said to exist, but neither a reference nor an analytical prove is given. I think one of the advantages of this approach with respect to a fully Bayesian approach is computational efficiency. A comparison of the computational times would be useful to compare the methods in the simulated and semi-simulated examples. A description of the organisation of the paper is missing at the end of the Introduction. Table 1: I am a bit surprised that the coverage of the proposed method is larger than other methods, but the size is similar of the credible intervals is smaller than other methods with low absolute error: the credible interval coverage should be similar to the frequentist level (since the credible intervals are of level 0.95, the coverage should be around 0.95), a larger coverage should mean larger intervals. I think a better explanation of the Table is needed. Line 103: saying that f, \pi, and m are independent under the posterior is imprecise, because they are conditionally independent (for example, they all depend on X). Line 106: since the posterior distribution on \psi depends on the full posterior distribution, which, I believe, is approximated by using full conditionals, I think you still need to define a prior for \pi, as it is also shown by Equation (4).

Reviewer 2



In their manuscript entitled, "Debiased Bayesian inference for average treatment effects", the authors present a new class of Bayesian model (or, as they phrase it, choice of [stochastic process] priors) for estimating average treatment effects (across a population) given only observational data not from an RCT (i.e., the canonical causal inference setting in social science or population health). The insight brought to this problem regards the structuring of the model to introduce a posterior de-biasing correction based on theoretical insights from the Bayesian asymptotics literature. From my point of view---having worked extensively with Gaussian process & Dirichlet process models and having read widely on the asymptotics of these processes---I was very pleasantly surprised to see here: (i) the identification of a connection between the ATE estimation problem in a semi-parametric Bayesian setting and this particular branch of the asymptotics literature, and (ii) that the authors were able to successfully transfer the insights from the asymptotics back to the practical problem to achieve a more effective model. Moreover, the presentation is very clear (modulo my concern that I find sometimes the exchangable use of the terms model and prior to be initially a little confusing). Although I found no errors in the text I felt that perhaps some discussion on how this model might interact with the (seemingly) increasingly common prior step of automatic causal feature selection could be warranted: e.g. presumably this method suffers from a curse of dimensionality if too many non-important variables are simply thrown into the design matrix, but likewise presumably performance will suffer if important covariates are omitted: is there any way within this model class to diagnose either of those problems?

Reviewer 3



** Update after author response** I'd like to thank the authors for their very detailed response to my and other referees' comments. In particular, I welcome the comparison to BCF and find it quite interesting that it performs better on the semi-synthetic data but not the synthetic! On re-reading the paper and supplement, as well as the response, I do think the way you have incorporated the propensity score is quite clever. I'm quite happy to revise my score up to 7. ----- The authors consider the important problem of heterogeneous treatment effect estimation. They specifically propose a non-parametric Bayesian procedure, placing a Gaussian process prior on the potential outcome function m(x,r). They note that the natural approach (i.e. placing a "vanilla" GP prior, e.g., on this function) can result in substantial bias in the estimate. The authors propose re-parametrizing m(x,r) to include an estimate of the propensity score and note that such a correction yields better performance than state-of-the-art methods. While I generally liked the approach of the paper, I cannot help but draw parallels to earlier work by Hahn and colleagues, which the authors have referenced. Indeed, Figure 1 displays a type of "regularization-induced confounding" described by this group in the context of linear models (reference 15 in the present manuscript) and tree-based methods (reference 16 in this paper). Moreover, the modified prior bears some similarity to the parametrization of m(x,r) used in reference 16, in that both decompose m into a term depending on an estimated propensity score and a term that does not include a propensity score estimate. I would note, however, that the proposed modification is arguably somewhat more principled than the one in reference 16 and I think some discussion comparing and contrasting these approaches is necessary. In a similar vein, I think a direct comparison with the Bayesian causal forest (bcf) method from reference 16 is warranted. Adding such a comparison should be straightforward, as the method is currently implemented in the "bcf" package available on CRAN. This comparison is perhaps more fair than simply running BART with the propensity score included as a covariate. Beyond this, I have a few additional concerns: - Estimation of F: I understand that to provide inference for a population average treatment effect, one needs to accurately model the distribution of X. As the authors note in passing, this can be quite difficult, especially when the covariates are high-dimensional and include both continuous and discrete outcomes. In general, posterior inference about the population ATE proceeds by repeatedly (i) drawing F* ~ F | D, a posterior sample of the distribution of X, (ii) X* ~ F*, a new set of covariates, (iii) computing m(X*,1) - m(X*,0). The authors propose using a Dirichlet Process prior on F with base measure equal to the empirical distribution of the observed covariates. The resulting posterior distribution of F places all of its probability mass on distributions that are supported only on the covariates observed in the sample. As a result, all inference about the population ATE is based on evaluating the potential outcome function on covariates already observed in the sample. This seems like a "population effect" only in the narrowest sense: it implicitly assumes that within the sample, one has observed all possible sets of covariates. -- Originality & Quality: To the best of my knowledge, the proposed representation of the function m(x,r) is new and I rather liked the motivation for it (provided in the supplement). It is especially promising that the proposed methods yields performance similar to the state-of-the-art. However, the general phenomenon described and overall approach bears striking similarity to existing work, which must be acknowledged. Clarity: The paper is well-written Significance: I do think the results are important and this paper opens up the possibility for better Bayesian non-parametric estimation of heterogeneous treatment effects.

[Author Response · NeurIPS 2019]

We thank the reviewers for their constructive suggestions and insightful comments. We have (1) added simulations for
Bayesian Causal Forests and (2) have substantially expanded the discussion in the final version to address the various
reviewer comments. A summary of the added discussion is provided point-by-point below.

**Reviewer 1: $\pi$ is not assigned a prior.** We use an empirical Bayes approach, which is, as the reviewer points out,
computationally much faster than a fully hierarchical Bayesian approach of placing a prior on $\pi$. For Gaussian
processes (GPs), since our approach reduces to modifying the prior covariance, the posterior can be computed using
standard computational tools for GPs. Conversely, one can only sample from the hierarchical Bayes posterior using a
Metropolis-Hastings-within-Gibbs-sampling algorithm, which is far slower in practice.

$\pi$ **requires a prior.** Please note that we consider *independent* priors on $(m, \pi, F)$ (for computational reasons), so that
the posterior also factorizes and hence the $\pi$ term is conditionally independent of $\psi$ (see lines 94-107 and Appendix A).

**Comment on Table 1.** The reason for our high coverage is that our posterior bias, due to our explicit ATE bias
correction (4), is (much) smaller than the posterior variance by design. As the reviewer correctly points out, asymptotic
theory predicts the frequentist coverage should converge to 0.95 as the sample size increases due to the semiparametric
Bernstein-von Mises theorem (cf Ray & van der Vaart (2018)). However, it is a subtle question as to when the asymptotic
regime applies and our examples seem insufficiently data rich for this to be the case (e.g. $n = 1000$ observations but
$d = 100$ input features).

**Missing organizational section/reference for bias/language precision.** We have incorporated these suggestions.

**Reviewer 2: Variable selection for causal inference.** We consider here a GP with squared exponential kernel with
automatic relevance determination (ARD), i.e. whose data-driven lengthscale $\ell_i$ represents the relevance of the *ith*
feature to the response surface. ARD has been used successfully for removing irrelevant inputs by several authors (see
e.g. Chapter 5.1 of Rasmussen & Williams (2006)) and can thus be viewed as a form of automatic (causal) feature
selection. Diagnosing missing significant covariates (confounders) is an important and difficult problem which requires
further investigation in the future.

**Reviewer 3: Comparison with Hahn et al.** We would firstly like to clarify that our goal is to improve estimation of
the *average treatment effect* (ATE) in the *presence* of heterogeneous treatment effects. It is known that naively using
product priors in casual inference/missing data problems can yield biased inference [this goes back to at least Robins
and Ritov (1997). The 'regularization-induced confounding' of Hahn et al. is a very nice illustration of similar ideas
for the concrete and important cases of linear models and BART priors]. One solution is to reparametrize to force the
missing information into the likelihood (e.g. Ritov et al. (2014), Hahn et al. (2018)), while another is to use propensity
score (PS) information (Rosenbaum & Rubin (1983)).

In a nice paper Hahn et al. (2017) successfully show that this latter idea also helps Bayesian estimation using BART.
Their approach is designed to improve nonparametric estimation of the *whole* response surface, which will also lead to
some improvement when estimating the ATE. However, it is known that even when the prior is perfectly calibrated (i.e.
all tuning parameters are set optimally) and recovers the entire response surface at the optimal rate, the posterior can
still induce a bias in the *marginal posterior* for the ATE $\psi$ that prevents efficient estimation and destroys uncertainty
quantification (see e.g. Ray & van der Vaart (2018)).

The specific form in which we include the PS in our prior (4) is very deliberate - it arises as the 'least favorable direction'
of the ATE in semiparametric statistical theory and is specifically designed for estimating the ATE. When either the PS
or response surface are especially difficult to estimate, we expect that incorporating the PS as a covariate as in Hahn et
al. (2017) will still induce a bias for the ATE (the theory in Ray & van der Vaart (2018) predicts this). In fairness, we
wish to emphasize that Hahn et al.'s goal is to estimate the *entire* response surface, for which they provide excellent
results, which is a different problem to estimating the ATE we consider here.

**Bayesian Causal Forest (BCF) simulations** have been added to the paper. In summary, for estimation BCF performs
well on the synthetic data (but moderately worse than our method in both the homogeneous and heterogeneous cases)
and excellently on the semi-synthetic data (moderately better than our method). For uncertainty quantification, BCF
typically had the shortest credible intervals with suboptimal coverage (80-85%) on the various synthetic datasets and
excellent coverage ($\sim$95%) on the semi-snythetic data.

**Estimation of $F$.** We use the widely used 'Bayesian bootstrap' (BB) since (1) it is computationally much faster (you
need only one costly Cholesky decomposition to generate posterior samples of the ATE whereas with the full Dirichlet
process (DP) posterior we require one per posterior draw) and (2) for moderate/large sample sizes it is very close to the
true DP posterior. We do not assume that 'one has observed all possible covariates', rather that our covariate samples
are representative of the population. If the observed covariates greatly differ from the underlying population distribution
then indeed this will not generalize well, but then neither will any prior not involving detailed outside expert information
for that particular application.

[Meta-Review · NeurIPS 2019]

Overall, the reviewers found this a valuable addition to the causal inference literature. While we would have liked to see more comparisons, we feel that that by incorporating the BCF simulations, and the clarifications mentioned in the rebuttal, this paper will be a welcome addition to the conference.